# Thermotolerance and Physiological Traits as Fast Tools to Heat Tolerance Selection in Experimental Sugarcane Genotypes

**Sergio Castro-Nava** [1],[*] and **Enrique López-Rubio** [2]

1. Universidad Autónoma de Tamaulipas, Facultad de Ingeniería y Ciencias, Cd. Victoria, Tamaulipas 87149, Mexico
2. Campo Experimental "El Estribo", Instituto Nacional de Investigaciones Forestales Agrícolas y Pecuarias (INIFAP), El Naranjo, San Luis Potosí 79310, Mexico; lore2010_montecillos@yahoo.com.mx
* Correspondence: scastro@docentes.uat.edu.mx; Tel.: +52-834-318-1721

**Abstract:** Heat stress limits the growth, development, and yield of crop plants when it occurs during short or long periods of time. An experiment was conducted with the objectives of (*i*) evaluating the cell membrane thermostability (CMT) as an indicator of heat tolerance in sugarcane and to determine its relationship with physiological parameters under heat-stressed conditions, and (*ii*) evaluating the utility of CMT for selecting heat-tolerant genotypes in a breeding program. A total of nine elite experimental, and four commercial sugarcane genotypes were evaluated for CMT, and the results are expressed as relative cell injury (RCI). Six genotypes were classified as highly tolerant and seven as highly sensitive. We concluded that the use of RCI, as an indicator of CMT in sugarcane genotypes, is a suitable useful parameter for selecting the genotypes tolerant to heat stress in a breeding program. This procedure, combined with other characters, helps to identify sugarcane plants with the ability to maintain a high yield photosynthetic rate under stressful field conditions. Furthermore, it offers an opportunity to improve selection efficiency over that of field testing, since high temperature stresses do not occur consistently under field conditions.

**Keywords:** sugarcane; heat stress; thermostability; relative cell injury; SPAD chlorophyll meter readings (SCMR); photosynthetic rate

## 1. Introduction

Occurrences of short, and perhaps long, periods of high temperature will be one of the most critical characteristics of future climates [1]. Climate models predict that global surface temperatures will continue to rise [2]. The ability of plants to neutralize stress conditions depends on the efficiency and speed at which they recognize the stress, generate signal molecules, and activate stress-protective mechanisms [3]. Heat stress, defined as a rise in temperature beyond the threshold levels sufficient to cause severe damage, substantially limits the growth and yield of crop plants when it occurs transiently or continually [4]. In other words, a high ambient temperature is a critical factor for plant growth and development. In general, a transient elevation in temperature, usually 10–15 °C above ambient, is considered as a heat shock or heat stress; transitory or consistently high temperatures cause a variety of morphological, physiological, and biochemical changes in plants, which affect plant growth and development and may lead to a drastic reduction in economic yield [5].

Heat tolerance is, therefore, an important breeding target in the genetic improvement programs for heat tolerance in sugarcane in México. This is also important because in genetic improvement programs for heat tolerance, the use of physiological parameters as selection criteria is very limited. The most

common approach is to evaluate the yield grown under heat stress in nurseries. Sugarcane breeders are seeking new heat-tolerant germplasm suited to the stressed areas in collaboration with our physiology program. Physiological and biochemical screening techniques can be a useful tool and could increase selection efficiency, and help in the selection of the desirable characteristics that can be lost during the selection process when using other criteria. Several heat stress-related traits have received considerable attention, in particular, electrolyte leakage measured by conductivity meters [6,7]. This procedure is based on the observation that when leaf tissue is injured by exposure to high temperature, cellular membrane permeability is increased, and electrolytes diffuse out of the cells. The electrolyte leakage test has been widely used to assess the level of plant tolerance to various stresses in plant species such as sugarcane [8–10]. The technique is relatively simple, repeatable, rapid, and requires inexpensive equipment.

The aims of the study were to (*i*) evaluate the cell membrane thermostability (CMT) as an indicator of heat tolerance in sugarcane and to determine its relationship with physiological parameters under heat-stressed conditions, and (*ii*) to evaluate the utility of CMT for selecting heat-tolerant genotypes in a breeding program.

## 2. Materials and Methods

A total of nine elite experimental sugarcane genotypes (EMex 02-59, EMex 05-222, EMex 00-80, EMex 03-198, EMex 05-141, EMex 05-225, EMex 01-323, EMex 02-05, EMex 00-82) and four commercial genotypes (Mex 95-60, Mex 95-52, Mex 95-59, Mex 95-03) were used for this study. The plant material was donated from the germplasm collection under evaluation at Campo Experimental "El Estribo", as a complementary collaborative work which classifies genotypes by their tolerance to drought and heat stress using physiological parameters. The experiment was carried out in 2018 and was sown on 15 September 2017. The genotypes were evaluated for their heat tolerance under high temperature in such a way that the plants were prevented from being under water stress through frequent watering during the biological cycle of all the genotypes, although this was not necessary during the rainy season. By the availability of genetic material, each genotype was planted in five meter rows spaced at 1.30 m in the nursery of the Facultad de Ingenieria y Ciencias, Universidad Autonoma de Tamaulipas, at Cd. Victoria, Tam. México.

The experimental material was evaluated for cell membrane thermostability (CMT) following the method proposed by Sullivan [6] with a modification of temperature treatment to 60 °C and for a duration of 60 min in a water bath (Boekel Grant BB-1400). The control vials were maintained in a water bath at ambient temperature (28 °C) during the same period. Castro [11] suggested that treatment temperature could be used to determine the treatment conditions that produce the greatest sensitivity for detecting real genetic differences. After temperature treatment, the vials were placed in an autoclave (Felisa Model FE-399) held at a pressure of 0.10 MPa and 120 °C for 10 min to completely kill plant tissue and release all of the electrolytes. As leaves of different ages might show differential response, CMT was measured on the second youngest fully expanded leaves at two phenological stages: (1) leaf development and (2) grand growth [12]. Samples collected at each phenological stage from each plot consisted of two sets (control and heat treated) of 10 leaf discs (10 mm in diameter) cut from 10 randomly selected plants within the row. The percentage relative cell injury (RCI %), as an indicator of CMT, was calculated with the following formula:

$$RCI\ (\%) = 1-\{[1-(T_1/T_2)]/[1-(C_1/C_2)]\} \times 100$$

where T and C refer to the electrical conductivity (EC) values of heat-treated and controlled vials, and subscripts 1 and 2 denote initial (before autoclaving) and final (after autoclaving) EC readings, respectively.

The physiological parameters were measured at the stage of grand growth on a completely sunny day (on day 28 in the supplemental Table S1) between 13:00 and 15:00, at air average temperature of

38.6 °C. The photosynthesis rate (A ($\mu$mol $CO_2$ m$^{-2}$ s$^{-1}$)), stomatal conductance (gs (mol m$^{-2}$ s$^{-1}$)), transpiration rate (E (mmol $H_2O$ m$^{-2}$ s$^{-1}$)) and leaf temperature ($T_{leaf}$ (°C)) were measured with a LI-6400 portable photosynthesis system (Li-COR, Lincoln, NE, USA) which was equipped with a $2 \times 3$ cm$^2$ LED chamber. We used a flow of 500 mol s$^{-1}$, a $CO_2$ concentration of 400 $\mu$mol mol$^{-1}$. The internal LED light source in the LI-COR 6400 was set at 2200 $\mu$mol m$^{-2}$ s$^{-1}$, so as to have a constant and uniform light across all measurements. The chlorophyll content was measured using a self-calibrating chlorophyll meter (SPAD 502, Minolta, Japan), after physiological measurements on the same day, in five active leaves (the average of each leaf on three positions: apex, middle, and base) of 10 plants of each genotype. The analysis of variance (ANOVA) was used to determine the differences among genotypes for the RCI and physiological parameters, using SAS [13]. The experiment for the RCI involved two factors (genotypes and phenological stage). To assess treatment effects, the genotype and phenological stages were considered as fixed effects and replicate block was considered a random effect. Combined analyses of variance were performed for each treatment using the SAS (Statistical Analysis System) GLM (General Lineal Model) procedure. The effects associated with the genotype and phenological stage, and their interactions, were identified. The data for physiological parameters at the grand growth stage were analyzed statistically by ANOVA. The experimental design was a randomized complete block design with 10 replicates. The standard error (SE) is shown as an estimate of variability, and the means of the physiological parameters were separated by Tukey's Studentized Range (HSD) Test at the $p = 0.05$ levels to provide separation of differences among the heat tolerant (HT) and heat sensitive (HS) groups.

## 3. Results and Discussion

RCI% is an indicator of cellular or tissue heat tolerance; low RCI% reflects high CMT and high RCI% low CMT [14]. The analysis of variance (data not shown) indicates that there is a significant difference between genotypes ($F_{12,100} = 2722$, $p \leq 0.01$) and the phenological stage ($F_{1100} = 352,380$, $p \leq 0.001$) in RCI (%). A significant genotype x phenological stage interaction ($F_{12,100} = 2002$, $p \leq 0.05$) for RCI (%) occurred, indicating that the genotypes performed differentially across the phenological stage for this variable. The relative cell injury (RCI) varied among 13 genotypes, with a range of 17%–75% and an average of 40.8%, when considering the phenological stage (Figure 1). These RCI values indicate a great difference in heat tolerance among the genotypes, as assessed by the CMT. Similar values of RCI% were observed in other experiments [7,9,15,16] on heat and drought stress in different crops.

The sugarcane genotypes responded differentially for the RCI% across the phenological stage. Regardless of the genotype, the impact of the phenological stage was very significant because the RCI% increased from 23.8 to 57.8% (34%). All genotypes increased the RCI at a different level when they went from the leaf development to grand growth. The exposure to high temperature (60 °C) in this study, during heat treatment in the CMT test, produced a better opportunity for discrimination between experimental genotypes and safely identified the genotypes that were heat-tolerant, especially at the grand growth stage.

For the response of the genotypes at the grand growth stage, the genotypes were separated into two groups or classifications of heat tolerance based on the RCI values, using the average obtained from all the genotypes at the phenological stage with the highest RCI (57.8%). Those genotypes producing RCI values less than or greater than that average were classified as heat tolerant (HT) or heat sensitive (HS), respectively. Accordingly, six genotypes (Table 1) were classified as HT (EMex 02-05, EMex 05-225, EMex 05-141, EMex 00-82, Mex 95-60, EMex 03-198), and seven as HS (EMex 05-222, EMex 02-59, EMex 00-80, Mex 95-03, Mex 95-52, EMex 01-323, Mex 95-59).

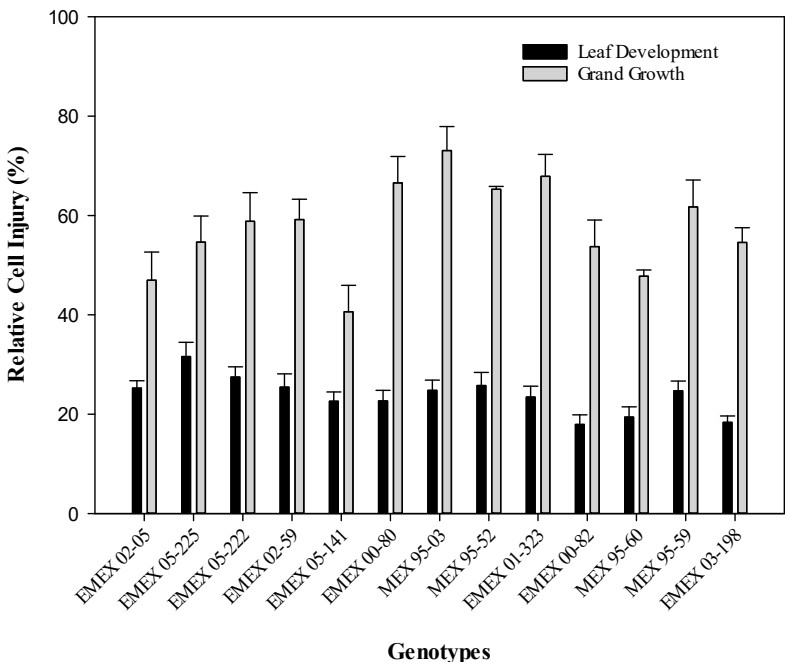

**Figure 1.** Relative cell injury (RCI) of 13 sugarcane genotypes grown under high temperature at two phenological stages, as determined by cell membrane thermostability (CMT) using a temperature of 60 °C for 60 min. Means ± SE (*n* = 10).

**Table 1.** Heat tolerance classification of nine sugarcane experimental elite genotypes and four commercial genotypes evaluated for relative cell injury (RCI), as an indicator of cell membrane thermostability under heat stress during the stage of grand growth.

| Classification | Number of Genotypes | Relative Cell Injury (%) | |
|---|---|---|---|
| | | **Mean** | **Range** |
| Heat-tolerant (HT) | 6 | 49.8 | 40–55 |
| Heat-sensitive (HS) | 7 | 64.7 | 58–74 |

Significant differences ($F_{12,36}$ = 2.472, $p \leq 0.05$) in A, gs, $T_{leaf}$, and SPAD occurred among the 13 genotypes (Table 2). The group of tolerant genotypes represents 46% of the total genotypes studied. Although part of the experimental genotypes (9), the tolerant ones represent 56%. The above shows that a large number of the experimental genotypes have heat tolerance due to a relatively low loss of electrolytes, as indicated by Thiaw and Hall [17].

**Table 2.** Analysis of variance for the physiological characteristics of the nine sugarcane experimental elite genotypes and four commercial genotypes, during the grand growth stage under heat stress conditions in field and to determine its relationship with the cell membrane thermostability as an indicator of heat tolerance.

| Source | d.f. | A ($\mu$mol $CO_2$ $m^{-2}$ $s^{-1}$) | gs (mmol $m^{-2}$ $s^{-1}$) | E (mmol $H_2O$ $m^{-2}$ $s^{-1}$) | $T_{leaf}$ (°C) | SPAD |
|---|---|---|---|---|---|---|
| | | Mean squares | | | | |
| Genotypes | 12 | 147.687 * ($p$ = 0.018) | 0.011 ** ($p$ = 0.009) | 8.120 ns ($p$ = 0.051) | 7.480 *** ($p$ = 0.000) | 50.004 ** ($p$ = 0.002) |
| Error | 36 | | | | | |
| CV (%) | | 30.8 | 16.3 | 33.4 | 1.52 | 9.8 |

d.f., degree of freedom; A, photosynthetic rate; gs, stomatal conductance; E, transpiration rate; $T_{leaf}$, leaf temperature; SPAD, soil plant analysis development. Ns non-significant; * significant at $p \leq 0.05$; ** significant at $p \leq 0.01$; *** significant at $p \leq 0.001$.

The group of HT genotypes had an 11% higher photosynthesis rate than the group of HS genotypes under heat (Figure 2A) due to a higher stomatal opening (Figure 2B), although with a lower total chlorophyll content (Figure 2C), but similar $T_{leaf}$ (Figure 2D). The genotype with the highest photosynthetic rate was Mex 95-60, a commercial genotype. This genotype exceeds by 2.2 times the average photosynthetic rate of the experimental genotype Emex 05-141 (Figure 3). However, the experimental genotype with the highest photosynthetic rate, and lower $T_{leaf}$ (Figure S1 in the Supporting Material) in the HT group was Emex 02-05 (28.55 $\mu$mol $CO_2$ m$^{-2}$ s$^{-1}$). In the HS group, it was found that the experimental genotype Emex 05-222 had a photosynthetic rate slightly higher (30.58 $\mu$mol $CO_2$ m$^{-2}$ s$^{-1}$) than Emex 02-05, which can be considered as an inconsistency. This inconsistency can be attributed to its genetic origin, since during its selection process, the selection had never been made using physiological criteria for its heat tolerance.

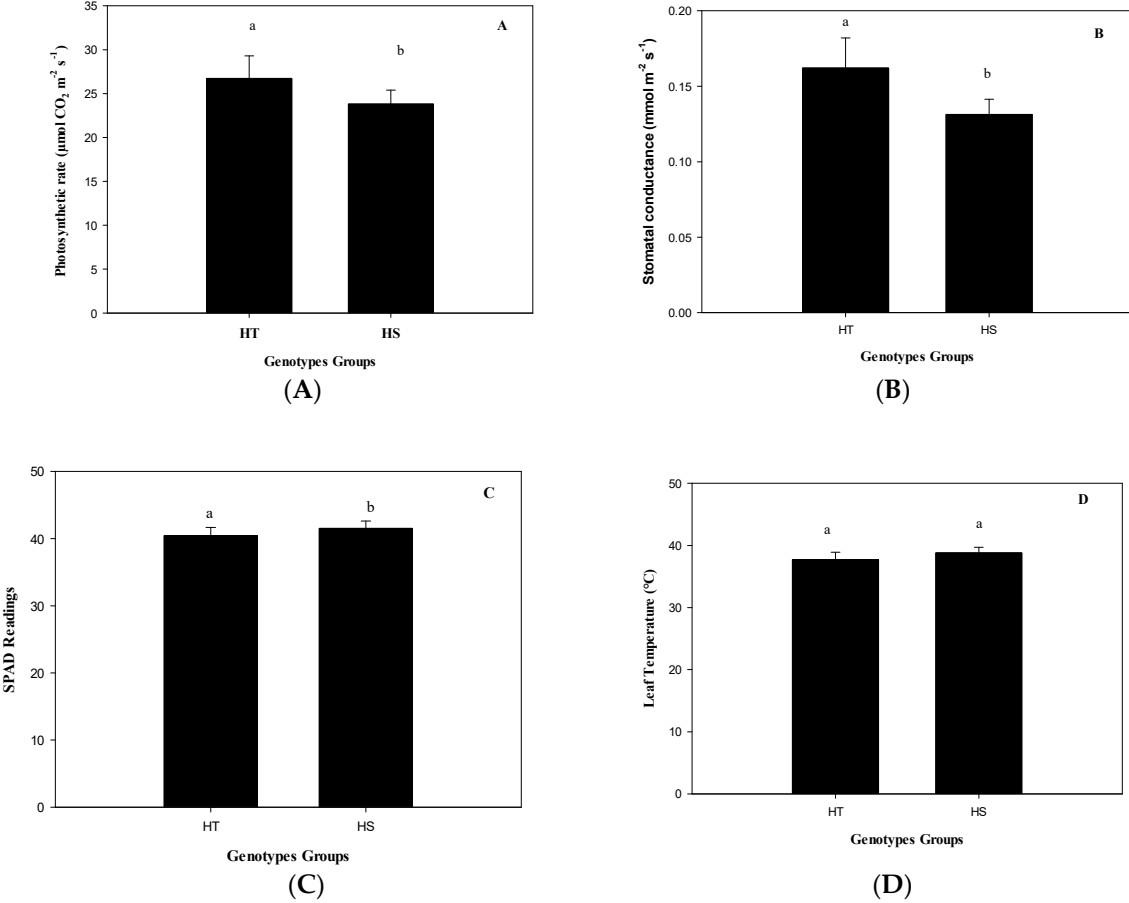

**Figure 2.** Photosynthetic rate (**A**), stomatal conductance (**B**), SPAD readings (**C**), and leaf temperature (**D**) of heat tolerant (HT), and heat sensitive (HS) sugarcane genotypes grown under high temperatures in the field during the 2017–2018 growing season. The different letters above the bars indicate statistically significant differences at $p < 0.05$ (Tukey test). Means ± SE ($n$ = 10).

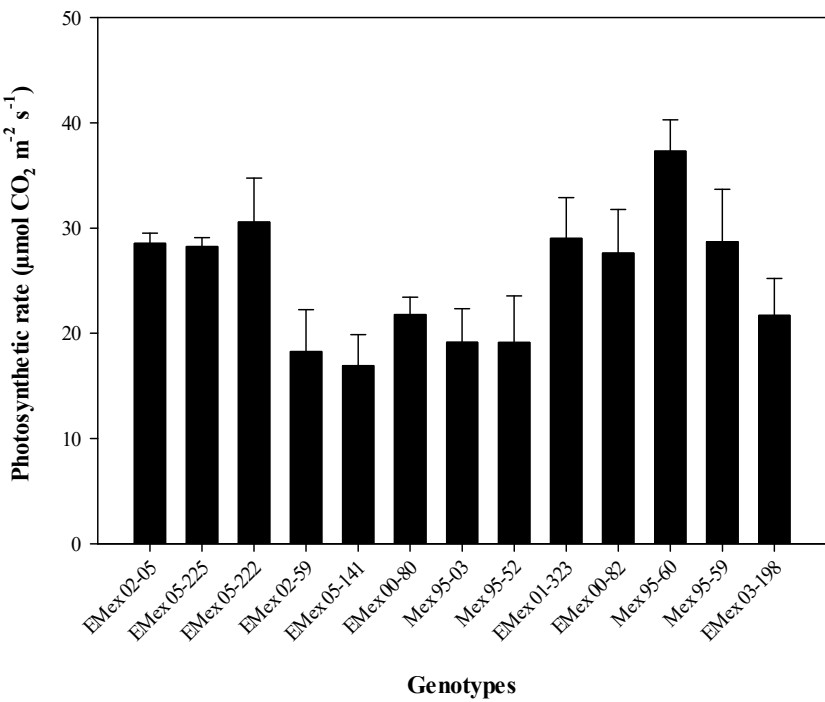

**Figure 3.** Photosynthetic rate of 13 sugarcane genotypes under high temperature in field, at the stage of grand growth during the 2017–2018 growing season. Means ± SE (*n* = 10).

According to the results obtained, it is possible to point out that the group of tolerant genotypes has greater thermotolerance of the cell membrane, as indicated by Rahman et al. [9] and Sudhakar et al. [14], which when associated with physiological processes such as the photosynthetic rate, can influence the yield in sugarcane. However, it is necessary to maintain the physiological selection criteria for heat tolerance in the following improvement process.

We concluded that the use of percentage relative cell injury (RCI %), as an indicator of the CMT in sugarcane genotypes, is a useful procedure to contribute to the selection of the sugarcane genotypes tolerant to heat stress in a breeding program, as Castro-Nava et al. indicated. [18]. This procedure, combined with other actors, helps to identify sugarcane plants with the ability to maintain high photosynthetic rate under stressful field conditions. Furthermore, this offers an opportunity to improve selection efficiency over that of field testing, since high temperature stresses do not occur consistently under field conditions.

**Supplementary Materials:** The following are available online at http://www.mdpi.com/2077-0472/9/12/251/s1, Figure S1: Leaf temperature of 13 sugarcane genotypes, Table S1: Weather conditions around the days of field measurements.

**Author Contributions:** S.C.-N. conceived, designed, performed the experiment, analyzed the data and wrote the paper; E.L.-R. contributed with the germplasm, revised this manuscript and give useful comments to improve this paper.

**Funding:** This research received no external funding.

**Conflicts of Interest:** The authors declare no conflict of interest.

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
