# Peer review of "Thermotolerance and Physiological Traits as Fast Tools to Heat Tolerance Selection in Experimental Sugarcane Genotypes"

_agriculture, doi:10.3390/agriculture9120251_

Round 1

Reviewer 1 Report

The revised manuscript has improved . All recommended corrections were made. However, it needs more grammar checking (for example the sentence in lines 40-41 is incomplete; in line 108 "data not shown" is correct).

Author Response

The corrections were made:

the sentence in lines 40-41 It's completed;

in line 108 "data not shown" was accepted.

Reviewer 2 Report

Response to Reviewer 1 Comments

Review of Castro-Nava & López-Rubio: „Use of thermotolerance and physiological parameters as fast tools to selection for heat tolerance in experimental sugarcane genotypes”; submitted to Agriculture

In their manuscript Castro-Nava and López-Rubio aim at establishing cell membrane thermostability as an indicator of heat tolerance in sugarcane. To this end, they measured ion leakage/conductivity of leaf discs from 13 distinct field-grown sugarcane genotypes after heat treatment at 60°C and compared the obtained relative cell injury (RCI) to an array of other physiological parameters that were measured directly in the field, i.e. photosynthetic rate, stomatal conductance chlorophyll content and leaf surface temperature. To evaluate whether RCI is suited to distinguish heat-tolerant from heat-sensitive genotypes, the authors divided the 13 cultivars into these two categories based on the observed RCI values and tested whether the two groups also differ in the other parameters measured. It was found that the six heat-tolerant genotypes showed higher photosynthetic rates, higher stomatal conductance and lower chlorophyll contents compared to the heat-sensitive genotypes. All parameters were measured around early afternoon on a sunny day, i.e. at elevated temperatures. The authors concluded that RCI, measured as ion leakage, can be used to identify sugarcane genotypes that possess an increased heat tolerance and thus maintain higher photosynthetic rates under increased ambient temperature.

The structure of the manuscript and the presentation of the results are quite clear with some minor exceptions only. The background presented in the introduction is sufficient to understand the aim of the study and all experiments presented. In my opinion, the conclusions are backed by the observations made, at least in the way they are presented here, and the authors do not try to make any overly speculative statements. Having said that, I will come to the main criticism concerning data analysis and conclusions made, though. Since the authors aim at establishing RCI as a predictor for heat tolerance, and the use thereof in breeding programs, I was wondering whether the conclusions would remain valid when a correlation analysis between the RCIs of the 13 genotypes and the photosynthetic rates was performed. In terms of predictive power such a negative correlation would represent a much stronger argument than the categorization into two groups and a rather coarse comparison between these. I would therefore appreciate such a correlation analysis, at least for RCI and the photosynthetic rates that were measured in the field.

Response:

According to the suggestion to perform a correlation analysis (RCI vs. photosynthetic rates), this was performed and a value of r = -0.21 (non-significant) was determined and a figure is attached as support material in which a trend line is added. However, we want to point out that the objective of ms is not to establish the RCI as a predictor of heat tolerance, rather the objective is to establish, as has been done in other crops, that it is a good tool in the process of selecting sugarcane genotypes heat tolerant. The simple fact of applying the RCI as a selection criterion is not recommended, due to the complexity of the character, it is also necessary to add other criteria. We know the complexity of the character, but the RCI can be useful in the process to rule out genotypes of a large number that is managed in an improvement program.

We do not agree that the comparison of genotypes in the groups is a coarse comparison, since doing so means testing and identifying heat tolerant genotypes in the same crop cycle. In addition, as part of the contribution of this ms, the test must be performed at a temperature of 60 °C in the grand growth stage.

In addition, while it is true that some of the genotypes had higher photosynthetic rates in the field it is impossible to say which of these genotypes are especially heat tolerant since the genotype-dependent differences in photosynthetic rates have not been measured under control conditions.

I suggest the authors to adjust their conclusions, especially in lines 151:

“We concluded that the use of percentage relative cell injury (RCI %), as indicator of CMT in sugarcane genotypes, is a suitable procedure for selecting genotypes tolerant to heat stress in a breeding program […]”, and address this issue actively and identify the limitation of the study design, respectively. In my opinion, it is safe, though, to conclude that RCI % might be used to identify such genotypes that show increased photosynthetic rates at elevated temperatures. To what extent this is due to a stronger heat tolerance is difficult to judge, as mentioned above, since a genotype with low photosynthetic rate under field conditions, i.e. at 38°C in this case, might only be little affected by temperature whereas one of the better performing genotypes might even have higher photosynthetic rates under optimum conditions. In that case, one would say that the latter one is more susceptible to heat.

Response:

In relation to measurements under control conditions, it should be noted that the same procedure includes a control when electrolyte readings are made after heat treatment at 60 °C and comparing the readings before (initial) and after (final) of making the step of the autoclave. If you want to test the effectiveness of the test, especially in the group of tolerant genotypes, it would be necessary to do another experiment for this purpose and this would take another cycle (at least 12 to 14 months). This test was carried out in corn by the first author and is supported by: Grain yield, photosynthesis and water relations in two contrasting maize landraces as affected by high temperature alone or in combination with drought. Maydica 59: 104-111; 2014.

For the suggestion of modifying the conclusion, it was modified.

Thank you very much for the clarifications and the modifications made.

Overall, I am satisfied by the modifications that have been made by the authors. I only have a few more remarks (see below).

You might also refer to the correlation analysis in your text and discuss to what extent this or the separation of the genotypes into two distinct groups is more informative for your aim. Only, I do not agree with the second to last sentence in your discussion (starting with. “This procedure predicts…”), since you do not evaluate whether RCI is truly able to predict higher photosynthetic rates in the field. There is a difference between the predictive power of RCI and the use as an indicator in combination with other parameters to identify heat tolerant genotypes. If RCI would be able to predict higher photosynthetic rates under stressful field conditions it should perform better than random drawing of potentially heat tolerant genotypes from the 13 genotypes you have used. Let’s assume that there are seven genotypes with high photosynthetic rates in the field based on Figure 3 (EMex 02-05, EMex 05-225, EMex 05-222, EMex 01-323, EMex 00-82, Mex 95-60, Mex 95-59) and the remaining six with low rates. Using RCI as an indicator resulted in the identification of six potentially heat tolerant genotypes, out of which four are in the group that displayed higher photosynthetic rates (EMex 02-05, EMex 05-225, EMex 00-82, EMex 95-60) and two are not. This might not slightly better compared to random drawing of six genotypes from Figure 3. But here, my statistics skills are not sufficient to suggest a test how to evaluate the predictive power. Therefore, I would suggest to phrase the respective conclusion a bit more carefully.

Additional remarks:

Abstract:

In my opinion you can delete highly and high, respectively, in line 19. “Tolerant” and “sensitive” genotypes would be sufficient. Also, strictly speaking you did actually not test “the suitability of CMT for selecting heat-tolerant genotypes in a breeding program” since you did not really examine the heat tolerance of your genotypes that were classified as tolerant (also applies to last paragraph of introduction).

Rather you might conclude that CMT might be useful in the identification of heat tolerant genotypes for further breeding, as you mention in the last paragraph of your discussion.

Results & Figures:

Table 2: The table’s heading is misleading since you did not establish the relationship between cell membrane thermostability and the physiological parameters you present in the table. Rather you determined whether the 13 genotypes differed statistically significantly with respect to these parameters.

Could you please also provide a graphic representation of the outcome of the leaf surface temperature measurements? If you feel like they do not really add to your study, please include them in the supporting material. I was wondering whether the genotypes that showed elevated stomatal conductance would also display lower leaf temperature due to increased transpiration rates. Even though this is not the scope of the paper for some readers this might represent an interesting information.

Response:

A figure is included with the leaf temperature of genotypes as graphic representation

Thanks, this is appreciated. Please add some measure of variation, either SD or SE to the bars in figure 2 of the respective file. Also, you should refer to this supporting file in your manuscript.

Considering the statistical analyses that have been carried out; please provide some additional information, i.e. F-value and degrees of freedom as well as the precise p-value. This concerns both the text body as well as table 2.

Response:

The suggested data were added in Table 2

Considering table 2: I am wondering whether you might have mixed some of the numbers. To me it does not seem as if you have given the correct F-ratio values, since an F-ratio value of 0.011 (see gs in table 2) should not give a statistically significant difference at α=0.05 with the respective degrees of freedom. Please double-check and make sure you give the correct values here.

Also, please include F values and degrees of freedom in the text body according to APA style (i.e. F12,36=147.687, p<0.001), e.g. in line 108 and following.

Figure 1. Error bars are missing. As I understood, the RCI measurements were done on ten replicates. Therefore, it should be possible to include some measure of within-group variation in the figure. Considering the figure cation, I would prefer some more information. The information provided should suffice to understand what has been done without having read the text, in my opinion. The authors might at least include the exact temperature that has been used during the treatment as well as the treatment duration. Also, number of replicates and some information on what is actually shown should be included, e.g. mean +/-SD, N=10.

Response:

Figure 1 was modified as suggested and the information requested was added.

Perfect! There is just some minor spelling error, e.g.it might rather read: “Relative cell injury (RCI) of 13 sugarcane genotypes grown under high temperature at two phenological stages as determined by cell membrane thermostability using a temperature of 60°C for 60 minutes. Means ± SE (n=10).

Figure 2. Here, please also provide some additional information in the figure caption. Especially, information on the number of replicates (10 plants?), mean+/-SE as well as the statistical test that has been performed and the level of significance that has been used. Here, I would also appreciate an additional sentence describing what has been done, e.g. that the parameters were measured directly on field-grown plants etc. This also applies to Figure 3.

Response:

The suggested information was added to Figure 2 and 3

Usually, the single figures of a composite figure are mentioned in the caption, e.g.: “Photosynthetic rate (A), stomatal conductance (B) and SPAD readings (C) of heat tolerant (HT) and heat sensitive (HS) sugarcane genotypes grown under high temperatures in the field during 2017/2018 growing season. Different letters above bars indicate statistically significant differences at p<0.05 (t-test). Means ± SE (n=10).”

Please, include the statistical test that has been done for figure 2 and mention the level of significance that is indicated by different letters above the bars.

Table 1. Please state which phenological stage the shown data belongs to. Also, I think that the text following the table is set incorrectly. Or does it belong to the table itself?

Response:

Table 1 added the suggested information and accommodated the indicated text.

Thanks.

Material & Methods:

Sampling: Were the leaf discs for RCI measurements at Grand growth taken at the same day as the physiological measurements? Please clarify.

Response:

It was clarified in the text

Thanks.

Does “rainfed” mean that there was no additional irrigation?

Response:

It was clarified in the text

Thanks for the clarification.

Related to the previous point: do you have any weather data for the period when your plants were growing? If so, could you please provide them in the supplementals? If not, maybe you could get them from a near-by weather station, if available. What are the weather conditions around the days of your sampling/field measurements? I am asking since drought might have a considerable effect on any of the parameters measured, including heat tolerance as such.

Response:

Clarifications were made in the text about “rainfed”. In addition, we must point out that the risk of drought did not happen, since frequent auxiliary irrigation was made to keep the humidity close to the field capacity during the crop cycle. For this reason the effect caused by the drought is ruled out, there was only the effect caused by high temperatures, which was the main objective of the work.

Thanks for the clarification.

Meteorological data are attached during physiological measurements.

Thanks for adding the data. You might refer to this table in your text, e.g. in line 85:

“…sunny day (one day after collecting the leaf samples, i.e. on day 28 in supplemental table 1) between…”

RCI measurements: in your formula you mention initial and final EC readings: does “initial” refer to EC directly after the 60 minutes of treatment and “final” to the maximum conductivity observed after boiling the samples? Please clarify!

Response:

In the text clarifies what initial and final means, in relation to the RCI formula.

Thanks!

Please state more clearly how the data from the physiological measurements were analysed. What was considered a biological replicate?

Response:

The relevant clarifications were made in the text regarding the analysis of physiological measurements.

This is appreciated, thanks!

Did you average the obtained values for one leaf (apex, middle, base) or did you average them across the five leaves from the same plant? Please clarify!

Response:

The equipment allows up to 30 measurements to be taken and the average obtained. Measurements were made on each sheet, as indicated (apex, middle and base), and the average was obtained and recorded and so on the five leaves of each plant and the average per plant of each genotype was obtained.

Thanks for the clarification.

Discussion:

In my opinion the discussion is concise but sufficient. The authors do not try to make any additional or speculative claims that are not based by the data presented. Please consider the point I raised in the second paragraph on the interpretation of the differences in photosynthetic rates in the absence of any control measurements, though. The authors might also develop a bit further on what is known between the relationship between photosynthetic rate and heat tolerance in other species.

Language/Spelling:

There are only few corrections needed concerning the language of the text.

In the abstract, please spell out CMT. Also upon the first mention in L52 of the introduction you should spell out the full meaning.

Response: corrected

In general, you may substitute all semi-colons for commas.

Response: Corrected in several cases

Sometimes ten is spelled out and sometimes it is given as number. Please unify.

Response: corrected

L43: I do not understand what you want to express with the very last part of the sentence “reason for this study”.

Response: corrected

L45: I do not understand the sentence on “heat tolerance genes […]”.

Response: corrected

L58: should read “was donated” instead of “were donated”

Response: corrected

L105: the second mentioning or the increase in percentage is misleading. It is also redundant since you already mentioned that RCI% increased from an average 23.8% to an average 57.8% from Leaf development stage to Grand growth stage.

Response: Modified

L127: “HT genotypes had 11% more photosynthesis […]”. This is rather imprecise language use. Please rephrase accordingly.

Response: Modified

Maybe rather: “… HT genotypes had an 11% higher photosynthetic rate than…”

Line 40: “Because the use of physiological…”  This sentence does not make any sense in the current form.

Author Response

Each and every one of the suggestions made to the manuscript was attended. In the attached file it is indicated in yellow what was corrected or what was modified, depending on the case. The most important modifications for the ms are indicated in green (figure 2).

This manuscript is a resubmission of an earlier submission. The following is a list of the peer review reports and author responses from that submission.

Round 1

Reviewer 1 Report

Review of Castro-Nava & López-Rubio: „Use of thermotolerance and physiological parameters as fast tools to selection for heat tolerance in experimental sugarcane genotypes”; submitted to Agriculture

In their manuscript Castro-Nava and López-Rubio aim at establishing cell membrane thermostability as an indicator of heat tolerance in sugarcane. To this end, they measured ion leakage/conductivity of leaf discs from 13 distinct field-grown sugarcane genotypes after heat treatment at 60°C and compared the obtained relative cell injury (RCI) to an array of other physiological parameters that were measured directly in the field, i.e. photosynthetic rate, stomatal conductance chlorophyll content and leaf surface temperature. To evaluate whether RCI is suited to distinguish heat-tolerant from heat-sensitive genotypes, the authors divided the 13 cultivars into these two categories based on the observed RCI values and tested whether the two groups also differ in the other parameters measured. It was found that the six heat-tolerant genotypes showed higher photosynthetic rates, higher stomatal conductance and lower chlorophyll contents compared to the heat-sensitive genotypes. All parameters were measured around early afternoon on a sunny day, i.e. at elevated temperatures. The authors concluded that RCI, measured as ion leakage, can be used to identify sugarcane genotypes that possess an increased heat tolerance and thus maintain higher photosynthetic rates under increased ambient temperature.

The structure of the manuscript and the presentation of the results are quite clear with some minor exceptions only. The background presented in the introduction is sufficient to understand the aim of the study and all experiments presented. In my opinion, the conclusions are backed by the observations made, at least in the way they are presented here, and the authors do not try to make any overly speculative statements. Having said that, I will come to the main criticism concerning data analysis and conclusions made, though. Since the authors aim at establishing RCI as a predictor for heat tolerance, and the use thereof in breeding programs, I was wondering whether the conclusions would reamain valid when a correlation analysis between the RCIs of the 13 genotypes and the photosynthetic rates was performed. In terms of predictive power such a negative correlation would represent a much stronger argument than the categorization into two groups and a rather coarse comparison between these. I would therefore appreciate such a correlation analysis, at least for RCI and the photosynthetic rates that were measured in the field.

In addition, while it is true that some of the genotypes had higher photosynthetic rates in the field it is impossible to say which of these genotypes are especially heat tolerant since the genotype-dependent differences in photosynthetic rates have not been measured under control conditions. I suggest the authors to adjust their conclusions, especially in lines 151: “We concluded that the use of percentage relative cell injury (RCI %), as indicator of CMT in sugarcane genotypes, is a suitable procedure for selecting genotypes tolerant to heat stress in a breeding program […]”, and address this issue actively and identify the limitation of the study design, respectively. In my opinion, it is safe, though, to conclude that RCI % might be used to identify such genotypes that show increased photosynthetic rates at elevated temperatures. To what extent this is due to a stronger heat tolerance is difficult to judge, as mentioned above, since a genotype with low photosynthetic rate under field conditions, i.e. at 38°C in this case, might only be little affected by temperature whereas one of the better performing genotypes might even have higher photosynthetic rates under optimum conditions. In that case, one would say that the latter one is more susceptible to heat.

Additional remarks:

Results & Figures:

Could you please also provide a graphic representation of the outcome of the leaf surface temperature measurements? If you feel like they do not really add to your study, please include them in the supporting material. I was wondering whether the genotypes that showed elevated stomatal conductance would also display lower leaf temperature due to increased transpiration rates. Even though this is not the scope of the paper for some readers this might represent an interesting information.

Considering the statistical analyses that have been carried out; please provide some additional information, i.e. F-value and degrees of freedom as well as the precise p-value. This concerns both the text body as well as table 2.

Figure 1. Error bars are missing. As I understood, the RCI measurements were done on ten replicates. Therefore, it should be possible to include some measure of within-group variation in the figure. Considering the figure cation, I would prefer some more information. The information provided should suffice to understand what has been done without having read the text, in my opinion. The authors might at least include the exact temperature that has been used during the treatment as well as the treatment duration. Also, number of replicates and some information on what is actually shown should be included, e.g. mean +/-SD, N=10.

Figure 2. Here, please also provide some additional information in the figure caption. Especially, information on the number of replicates (10 plants?), mean+/-SE as well as the statistical test that has been performed and the level of significance that has been used. Here, I would also appreciate an additional sentence describing what has been done, e.g. that the parameters were measured directly on field-grown plants etc. This also applies to Figure 3.

Table 1. Please state which phenological stage the shown data belongs to. Also, I think that the text following the table is set incorrectly. Or does it belong to the table itself?

Material & Methods:

Sampling: Were the leaf discs for RCI measurements at Grand growth taken at the same day as the physiological measurements? Please clarify. Does “rainfed” mean that there was no additional irrigation? Related to the previous point: do you have any weather data for the period when your plants were growing? If so, could you please provide them in the supplementals? If not, maybe you could get them from a near-by weather station, if available. What are the weather conditions around the days of your sampling/field measurements? I am asking since drought might have a considerable effect on any of the parameters measured, including heat tolerance as such. RCI measurements: in your formula you mention initial and final EC readings: does “initial” refer to EC directly after the 60 minutes of treatment and “final” to the maximum conductivity observed after boiling the samples? Please clarify! Please state more clearly how the data from the physiological measurements were analysed. What was considered a biological replicate? Did you average the obtained values for one leaf (apex, middle, base) or did you average them across the five leaves from the same plant? Please clarify!

Discussion:

In my opinion the discussion is concise but sufficient. The authors do not try to make any additional or speculative claims that are not based by the data presented. Please consider the point I raised in the second paragraph on the interpretation of the differences in photosynthetic rates in the absence of any control measurements, though. The authors might also develop a bit further on what is known between the relationship between photosynthetic rate and heat tolerance in other species.

Language/Spelling:

There are only few corrections needed concerning the language of the text.

In the abstract, please spell out CMT. Also upon the first mention in L52 of the introduction you should spell out the full meaning.

In general, you may substitute all semi-colons for commas.

Sometimes ten is spelled out and sometimes it is given as number. Please unify.

L43: I do not understand what you want to express with the very last part of the sentence “reason for this study”.

L45: I do not understand the sentence on “heat tolerance genes […]”.

L58: should read “was donated” instead of “were donated”

L105: the second mentioning or the increase in percentage is misleading. It is also redundant since you already mentioned that RCI% increased from an average 23.8% to an average 57.8% from Leaf development stage to Grand growth stage.

L127: “HT genotypes had 11% more photosynthesis […]”. This is rather imprecise language use. Please rephrase accordingly.

Reviewer 2 Report

The manuscript is well drafted but needs some modifications.

In the abstract, you use the abbreviation CMT without explaining what is means.

Line 19: it should be "highly" instead of "high"

Line 22: "furthermore it offers" is correct

Line 26: comma should be a semi-colon

The first paragraph of the introduction needs re-structuring. It seems a collection of statements without any clear structure. Please re-write it.

Line 40: is this true only for Mexico? Or is it especially relevant for Mexico? Why? Please explain it.

Lines 46-49: What is the basic principle of electrolyte leakage and why is it relevant?

The figure legends are missing. What is shown? The average with standard deviation? What is the number of replicates? Which statistical test did you use?

The classification of genotypes into heat-tolerant and heat-sensitive ones is not clear. Why did you choose the RCI for it? Looking at the RC ranges in table 1, they are very closely together (heat tolerant with RCI up to 55% and heat sensitive with RCI from 58%). Using the limit of 57,8% seems to be rather randomly. Please explain your choice further.

Check the references. Some doi are underlined.
